# Intensification of the Extraction Yield of *Eucalyptus globulus* Phenolic Compounds with Pulsed Electric Field

**Manel Nardjes Toumi [1], Abdelfettah Benyamina [1], Mohamed Ali Bouzidi [1], Abdelkader Semmak [2], Yassine Bellebna [2], Fawzia Toumi [1] and Amar Tilmatine [2,*]**

1   LEDE Laboratory, Djillali Liabes University of Sidi Bel-Abbes, Sidi Bel Abbes 22000, Algeria
2   APELEC Laboratory, Djillali Liabes University of Sidi Bel-Abbes, Sidi Bel Abbes 22000, Algeria
*   Correspondence: amar_tilmatine@ieee.org

**Abstract:** *Eucalyptus* extract-based pharmaceutical products are widely used because of their medicinal properties and their rich content of secondary metabolites, mainly phenolic compounds. This study aimed to maximise the extraction yield of these compounds and reduce the extraction duration by using a pulsed electric field (PEF) level of 6 kV/cm. The pulse width (T), number of pulses (n), and solvent concentration [C] were analysed. Several ethanolic extracts were obtained from the leaves of *Eucalyptus globulus*, and the content of total phenols, total flavonoids, and condensed tannins was measured through spectrophotometry. The results, obtained immediately after PEF treatment, revealed that for optimal values of the analysed factors, the total phenol content doubled and the flavonoid content increased significantly. However, PEF pre-treatment had no effect on the tannin yield. Moreover, optimisation was performed using the design of experiments methodology for identifying optimal values of the analysed factors.

**Keywords:** *Eucalyptus globulus*; pulsed electric field; total phenols; flavonoids; condensed tannins



## 1. Introduction

Since ancient times, the bark and leaves of various *Eucalyptus* species have been used as folk medicines for the treatment of diseases such as cold, fever, toothache, diarrhoea, and snake bites [1]. Approximately 500 *Eucalyptus* species are used to extract essential oils that contain many types of terpene compounds, including the popular 'eucalyptol', which is being used in medicine and perfumery [2].

Although phytochemical studies on *Eucalyptus* tend to focus on essential oils, many triterpenoids, flavonoids, tannins, and other non-volatile compounds have been isolated from this plant. Pharmacological studies have revealed that these compounds possess numerous biological antioxidants, as well as antiviral, antifungal, and antibacterial activities [1,3,4]. Ellagic acid, a phenolic compound isolated from the *Eucalyptus* bark, showed a stronger antioxidant activity than tocopherol (vitamin E) [5]. Other phenolic compounds with strong antioxidant activities such as tannins and acylated flavonol glycosides are all isolated from *Eucalyptus* [6]. The ease of cultivation and the rapid growth of this species make it a valuable natural resource for the commercial production of pharmaceutical products. Today, the benefits of *Eucalyptus* are well recognised [7,8]. In addition to its essential oils, *Eucalyptus* extracts are standardised and prepared in several galenic forms and are part of the arsenal of medicines available for the treatment or prevention of several human diseases.

The medicinal value of *Eucalyptus* extracts confirmed in studies has made yield optimisation of active ingredients in general and phenolic compounds in particular an interesting research topic. In fact, several studies have been conducted using solvent variation, supercritical fluids, microwaves, and ultrasound technology [9–12].

Phenolic compound extraction requires a mass transfer from the plant material. Different pre-treatment methods can be employed to achieve electroporation of the tissue

structure and thus improve the release of phenolic compounds. In addition to mechanical and enzymatic treatments, the use of pulsed electric fields (PEFs) is a highly effective method for the pre-treatment of food products in an eco-friendly manner [13–16]. PEF is generally used to induce electroporation of biological membranes, which facilitates diffusion in the tissues and mass transfer with the surrounding environment. Many studies have shown that the extraction yield obtained from apple and grape tissues can be improved by using PEF [17–20].

The electroporation mechanism of biological membranes is based on a high-intensity external electrical field. PEF is considered as an emerging nonthermal food processing system which is attracting increasing attention because of its capability for liquid food pasteurization and to enhance conventional processing operations by application of microsecond duration high-voltage electric potential to materials located between two parallel electrodes. PEF causes a transmembrane electric potential difference at the extremities of the cell membrane and if its level reaches a critical value (breakdown potential), electroporation of the membrane occurs and cell permeabilization increases causing the improvement of the mass transfer such as extraction of plant metabolites [21–23].

The electroporation (formation of pores) causes the increase of the membrane permeability inducing thus either the inactivation of microbes or enhancing the transfer of solutes via cell membranes. The action of PEF is based on a non-thermal effect, and in contrast to other non-thermal methods, the PEF technique can be applied in continuous flow treatment and needs short processing time. Note that solid–liquid separation is an operation widely used in the food industry, that consists in a mass transfer of determined compounds located inside the cells and are drifted to the liquid phase. Therefore, extraction efficiency is strongly dependant on the permeability level of the cell membranes [24–26].

This study investigated the applicability of PEF for the extraction of total phenols, total flavonoids, and condensed tannins from *Eucalyptus globulus* leaves, with a special focus on yield enhancements and a reduction in processing time. In this context, the effects of the pulse number, pulse width, and ethanol concentration were analysed using response surface methodology. The data were then modelled to optimise the process.

## 2. Materials and Methods

### 2.1. Material Sampling and Preparation

*E. globulus* leaves were collected from a tree cultivated at the Faculty of Natural and Life Sciences (Djillali Liabes University, Algeria). The plant material was first compared with the two specimens of *Eucalyptus globulus* deposited in the herbarium of the Missouri Botanical Garden (USA) under the references: MO2313003 and MO3393722 and then confirmed by Professor Bouzidi Mohamed Ali, who is expert in the fields of phytosociology, botany, and biogeography. Before starting the experiments, the leaves were immediately dried in an oven at 55 °C for 3 h and cut into almost square-shaped, small pieces with an average size of approximately 5 mm. All extractions in this study were performed within a solvent composed of ethanol and water using a plant/solvent ratio of 10%. The extraction time, ranging on an average between 2 and 7 min, was equivalent to the period during which the sample was subjected to electric field pulses; the extract obtained immediately after PEF treatment was recovered in a tube and then analysed.

### 2.2. PEF Equipment

During extraction, the samples were treated with PEF using a pilot scale system (EPULSUS-10, EnergyPulse Systems, Lisbon, Portugal) comprising a pulse generator and a treatment chamber (Figure 1). The PEF generator is characterised by a peak voltage of 10 kV with a maximum average power of 3 kW, and a repetition rate of 1–200 Hz. The treatment chamber comprises two parallel stainless steel disk electrodes with a diameter of 5 cm and placed at a gap of 0.5 cm. The crushed *Eucalyptus* leaves were placed in the chamber with 10 mL of solvent to improve the conductance with an electric field strength

equal to 6 kV/cm. After completing the PEF treatment, the product was removed and the phenolic compound content was immediately determined.

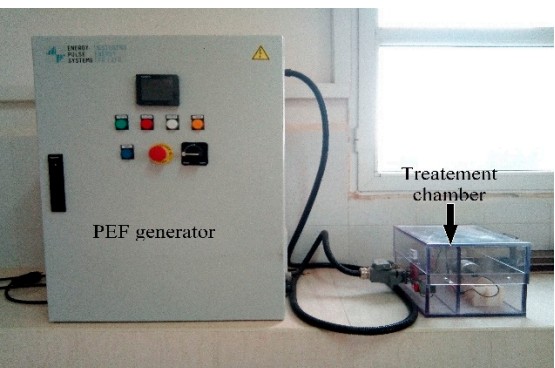

**Figure 1.** A photograph of the PEF generator and the treatment chamber.

### 2.3. Determination of Phenolic Compound Content

2.3.1. Total Phenolic Content

Total phenol content was determined according to the method described by Boizot and Charpentier [27]. A volume of 200 µL of the extracts was put into test tubes, and the mixture of 1000 µL of the 10-fold diluted Folin–Ciocalteu reagent and 800 µL of sodium carbonate (75g/L) was added. Then, the tubes were shaken and kept for 30 min at room temperature. The absorbance was measured at 765 nm by using a spectrophotometer against a blank that did not contain the extract. The calibration curve was run in parallel under the same operating conditions by using gallic acid (0–200 µg/mL) as a positive control. The obtained results are indicated as milligrams (mg) equivalent of gallic acid per 100 g of dried leaves (mg GAE/100 g dried leaves).

2.3.2. Total Flavonoid Content

Flavonoids were quantified using the method adopted by Zhishen et al. [28]. The method involved mixing 500 µL of the extract with 1500 µL of distilled water. Then, 150 µL of 5% sodium nitrite was added to the mixture, which was then mixed for 5 min. Next, 150 µL of 10% aluminium trichloride was added and mixed for 6 min. Finally, 500 µL of 4% sodium hydroxide was added. The absorbance value of the solution was then recorded at 510 nm. The results of this measurement were expressed as milligrams equivalent of catechin per 100 g of dry leaves, or mg CE/g 100 g dried leaves. Using catechin (0–200 µg/mL) as a positive control, a calibration curve was constructed simultaneously under the same experimental conditions.

2.3.3. Condensed Tannin Content

Condensed tannin content was determined using the vanillin method in acid medium [29]. A volume of 50 µL of the extracts was introduced in 1500 µL of vanillin/methanol solution (4%, *w/v*) and then mixed through vortexing. After that, a volume of 750 µL of concentrated hydrochloric acid was added. The resulting mixture was let to react at room temperature for a duration of 20 min. The measurement of the absorbance was performed at 550 nm against a blank by using a spectrophotometer. The calibration curve was run simultaneously under the same experimental conditions by using catechin (0–200 µg/mL) as a positive control. The obtained results are expressed as milligram (mg) equivalent of catechin per 100 g of dried leaves (mg CE/g 100 g dried leaves).

### 2.4. Experimental Design Methodology

Design of experiments is generally used for the three following successive steps: screening; optimization; and robustness testing [30–33]. The first stage of the experimental procedure is to carry out preliminary screening experiments which are employed at the

beginning of the procedure and are applied to identify the appropriate ranges of the studied factors. The screening experiments are carried out in this paper to determine the variation domain of the following three factors that can be easily adjusted.

- Pulse width T (μs).
- Number of pulses n.
- Concentration of ethanol [C] (%).

When the study case is a relatively simple process as the extraction of the phenol substances, classical "one-factor-at-a-time" experiments are considered to be more suitable than other types of design. Once the variation domains of each factor are determined, the optimization stage is then applied for identification of the "set point," for which the values of the analyzed factors lead to the optimal response of the process that could be maximum, minimum, or close to a target. In the present work, the response is the maximization of the phenolic compound extraction rates. The last stage to be carried out is the robustness testing to make certain that the response is not easily affected by small modifications of the values around the set optimal point. If the robustness is not achieved, the factors should be regulated for the outcome to remain within given specifications.

### 2.5. Extraction Kinetics

The extraction kinetics was studied according to time in order to determine the required time to extract the maximum amount of phenolic compounds. For this purpose, samples of *Eucalyptus* leaves treated with PEF (optimal conditions obtained by response surface modelling) were left to macerate for 24 h. Samples were then collected after 0.5, 1, 2, 4, 8, 16, and 24 h for assays.

### 2.6. Statistical Analyses

Four (4) replicas were performed for each plotted point, in all figures except the results summarized in Table 1 representing the results of a CCF experimental design. Results are presented as mean ± standard deviation. Comparison of means was performed by the one-way analysis of variance (ANOVA) followed by a multiple comparison by Tukey's test. Differences were considered significant when the *p* value was less than or equal to 0.05. Statistical analyses were performed with Minitab 20.3 statistical software (Minitab, Coventry, UK).

**Table 1.** Results of the CCF design experiments.

| Exp. No | Pulse Width T (μs) | Pulse Number n | Ethanol Concentration [C] (%) | TPC mg/100 g | TFC mg/100 g | CTC mg/100 g |
|---|---|---|---|---|---|---|
| 1 | 25 | 200 | 20 | 463.4 | 241.3 | 506.3 |
| 2 | 75 | 200 | 20 | 545.5 | 285.1 | 492.5 |
| 3 | 25 | 400 | 20 | 663.8 | 347.8 | 556.3 |
| 4 | 75 | 400 | 20 | 529.2 | 259.3 | 495.0 |
| 5 | 25 | 200 | 60 | 597.2 | 348.3 | 507.5 |
| 6 | 75 | 200 | 60 | 454.4 | 247.4 | 416.3 |
| 7 | 25 | 400 | 60 | 615.4 | 358.3 | 466.3 |
| 8 | 75 | 400 | 60 | 658.9 | 379.1 | 475.0 |
| 9 | 25 | 300 | 40 | 543.5 | 278.8 | 417.5 |
| 10 | 75 | 300 | 40 | 503.1 | 261.7 | 396.3 |
| 11 | 50 | 200 | 40 | 422.0 | 215.1 | 451.3 |
| 12 | 50 | 400 | 40 | 519.6 | 266.5 | 556.3 |
| 13 | 50 | 300 | 20 | 479.1 | 245.3 | 521.3 |
| 14 | 50 | 300 | 60 | 535.7 | 299.7 | 562.5 |
| 15 | 50 | 300 | 40 | 596.1 | 312.3 | 576.3 |
| 16 | 50 | 300 | 40 | 615.4 | 311.1 | 586.3 |
| 17 | 50 | 300 | 40 | 589.5 | 300.9 | 581.3 |

## 3. Results and Discussion

In this study, the experiments were conducted by considering three factors: pulse width T (μs), number of pulses (n), and ethanol concentration [C] (%). The experimental design methodology is generally used for screening, optimisation, and robustness testing. Screening experiments, also known as 'one-factor-at-a-time' experiments, were designed to identify the domain of variation of variables. The optimisation stage should give factor values for which the phenolic compound yield is maximum.

### 3.1. Screening Experiments

Variation limits of the three analysed factors were defined by conducting 'one-factor-at-a-time' experiments for a constant applied voltage (V) of 3 kV, corresponding to a constant electric field level of 6 kV/cm.

Experiment 1.1.: Variable pulse width T (2–150 μs) at constant values of the number of pulses n = 200 and ethanol concentration [C] = 50%.

Experiment 1.2.: Variable pulse number n (100–400) at constant values of pulse width T = 100 μs and ethanol concentration [C] = 50%.

Experiment 1.3.: Variable ethanol concentration [C] (0–60%) at constant values of the number of pulses n = 200 and pulse width T = 100 μs.

The results obtained in the aforementioned experiments are presented in Figures 2–4. The extraction yields of total phenol content, total flavonoid content, and condensed tannin content were selected for the evaluation of PEF pre-treatment and represented as a function of the three control factors.

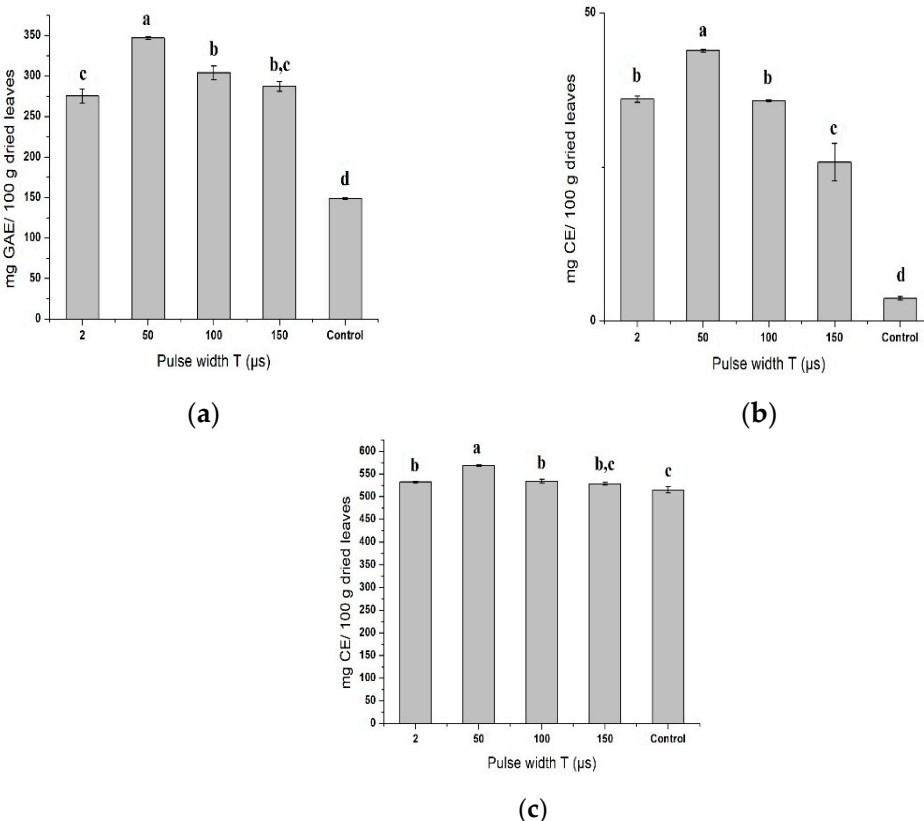

**Figure 2.** Variation of the phenolic compound extraction yield as a function of the pulse width. (**a**) Total phenolic content; (**b**) Total flavonoid content; (**c**) Condensed tannin content. Means not sharing any letters are significantly different.

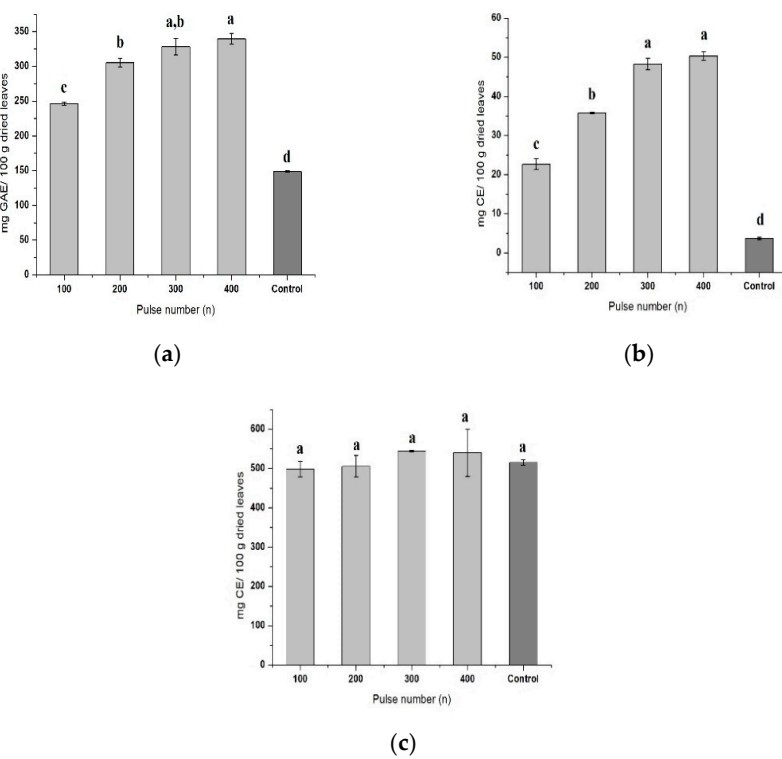

**Figure 3.** Variation of the phenolic compound extraction yield as a function of the pulse number. (**a**)Total phenolic content; (**b**)Total flavonoid content; (**c**) Condensed tannin content. Means not sharing any letters are significantly different.

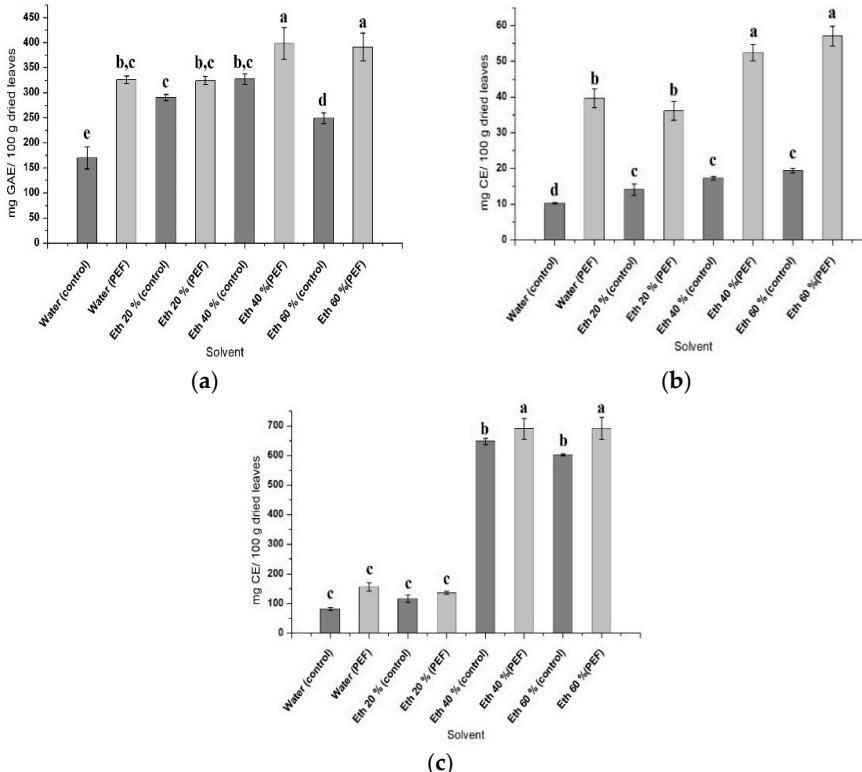

**Figure 4.** Variation of the phenolic compound extraction yield as a function of the ethanol concentration. (**a**) Total phenolic content; (**b**) Total flavonoid content; (**c**) Condensed tannin content. Means not sharing any letters are significantly different.

### 3.2. Set Point Identification

The optimal values of control variables were determined by using a central composite face-centered (CCF) design; the two limits 'max' and 'min' of the three factors $(T_{min}, T_{max})$, $(n_{min}, n_{max})$, and $(C_{min}, C_{max})$ are established in the previous section and are presented below.

$$T_c = (T_{min} + T_{max})/2 = \frac{25 + 75}{2} = 50 \text{ μs} \tag{1}$$

$$n_c = (n_{min} + n_{max})/2 = \frac{200 + 400}{2} = 300 \tag{2}$$

$$C_c = (C_{min} + C_{max})/2 = \frac{20 + 60}{2} = 40\% \tag{3}$$

The results of the CCF design experiments are presented in Table 1.

MODDE 5.0 software (Umetrics, Stockholm, Sweden), that is a Windows programme dedicated to the experimental designs, was used [34]. The programme calculates mathematical model coefficients and identifies the best adjustments of the factors to optimise the response. Moreover, the programme estimates two important statistical criteria that determine whether the mathematical model is valid, symbolised as $R^2$ and $Q^2$. To validate a mathematical model, both criteria should be equal or very close to the unit.

The mathematical models of the three responses, namely the extraction yield values of total phenol content, total flavonoid content, and condensed tannin content, were given by MODDE 5.0 and plotted, as shown in Figure 5. Because both criteria $R^2$ and $Q^2$ were very close to the unit, the three models were validated and therefore used for optimisation and prediction analyses.

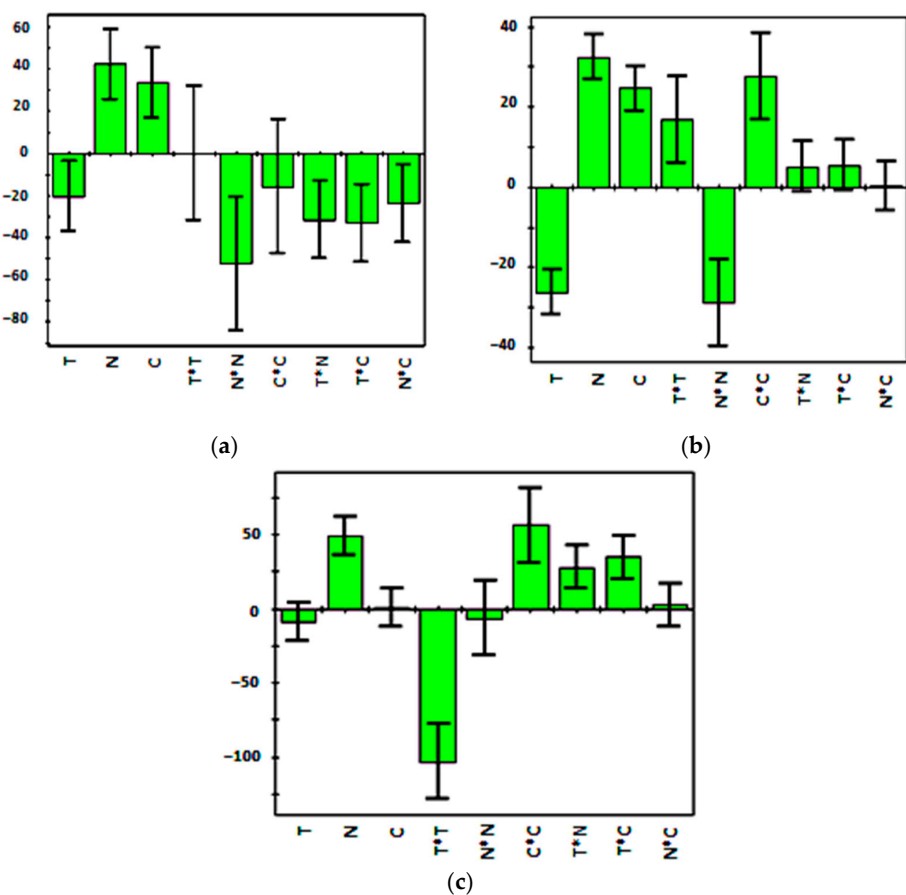

**Figure 5.** Coefficients of mathematical models plotted by MODDE. (**a**) Total phenols; (**b**) Total flavonoids; (**c**) Condensed tannins.

According to the obtained mathematical models (Figure 5), the coefficients of the three factors had an almost similar effect on the amount of phenol and flavonoid extracted. The effect of pulse width was negative, whereas the effects of pulse number and ethanol concentration were positive. Therefore, the phenol and flavonoid amounts decreased with the pulse width and increased with both pulse number and ethanol concentration. These results remain valid only within the variation domain of each factor defined previously. As observed in the results, the disruption of cell membranes increased with an increase in the number of pulses. The increased electroporation of cell membranes leads to the transfer of intracellular components from the disrupted cells [35]. Submitting the plant tissues in the PEF increased cell wall permeability and thus the extraction of intercellular products. Low-energy PEF for a short period results in the electroporation of cell membranes with minimum deterioration of the quality of plant compounds extracted [36]. Electroporation of the cell membrane by PEF causes a high extraction yield of the phenolic compounds. When subjected to a PEF, a transmembrane electric voltage is thus created on the extremities of the cell membrane thereby causing the separation of molecules on the charge mass basis [37].

On the other hand, for tannins, the coefficients associated with factors T and [C] were nonsignificant; tannins were indeed little influenced by PEF.

### 3.3. Optimisation Step

The software MODDE.05 allows to identify the optimal values of the factors that would generate the highest yield of phenolic compounds. It contains an optimisation routine that can analyse several responses simultaneously and which are affected by different weighting coefficients (Table 2). Given to the three obtained models, the process (i.e., maximising the phenolic compound yield of the three compounds simultaneously) is optimised for T = 39 μs, n = 356, and [C] = 60%, corresponding to extraction amounts of 582, 340, and 568 mg for phenol, flavonoids, and tannins, respectively (Table 3). To the best of our knowledge, PEF has never been applied to extract phenolic compounds from *Eucalyptus*. However, other optimization techniques have been used. The comparison of our results with those of other authors is relatively unfair because the content of phenolic compounds is variable depending on the season of harvest, the age of the plant, and the environmental conditions. It is more accurate to compare the effectiveness of the optimization technique used versus the control (conventional ex-traction) in the same work. Our results revealed that treating the plant with PEF for a very short extraction time increases the phenolic compound content approximately by 94%, 2166 %, and 9% for TPC, TFC, and CTC compared to the best yields obtained by simple maceration with ethanol. Gullón et al. (2017) performed a study on the influence of temperature, time, and ethanol/water ratio, to analyse their effects on extracting phenolic compounds from Eucalyptus globulus leaves using the response surface methodology [9]. Their results revealed a 5% increase for TPC and a 10% increase for TFC. Another study of optimization of phenolic compounds from another Eucalyptus species (*E. robuta*) by microwave-assisted extraction showed a 100% increase in yield for TPC and TFC compared to the lowest levels obtained in this same study [11]. The microwave-assisted extraction seems to be more efficient than other optimization techniques applied on this species in other studies, such as the use of ultrasound and supercritical fluids [10,12].

**Table 2.** Presentation of the optimisation routine of MODDE 5.0 used for maximising the phenolic compound yield of the three compounds simultaneously.

|   | Response | Criteria | Weight | Min | Target |
|---|----------|----------|--------|-----|--------|
| 1 | Phenol | Maximize | 1 | 594.433 | 622.503 |
| 2 | Flavonoid | Maximize | 1 | 348.083 | 364.758 |
| 3 | Tanins | Maximise | 1 | 568.68 | 591.68 |

Note that 'iter' and log(D) are the iterations number and the log of overall distance to the target, respectively; log(D) is equal to zero when all the responses are lying between Target and Limit values [34].

**Table 3.** Results of the optimisation routine.

|  | Pulse with | Pulses Number | Ethanol Concentration | Phenol | Flavonoid | Tanins | Iter | Log(D) |
|---|---|---|---|---|---|---|---|---|
| 1 | 31.7183 | 373.281 | 20 | 518.11 | 308.153 | 570.436 | 5001 | 0.9413 |
| 2 | 25.0017 | 370.821 | 20 | 520.402 | 323.891 | 528.312 | 5000 | 0.9515 |
| 3 | 33.7269 | 355.863 | 60 | 597.531 | 348.11 | 536.394 | 5000 | 0.4018 |
| 4 | 74.9998 | 400 | 60 | 438.3 | 324.987 | 575.681 | 5000 | 1.2152 |
| 5 | 38.7139 | 360.684 | 60 | 581.963 | 340.647 | 568.918 | 5000 | 0.2352 |
| 6 | 32.7857 | 369.125 | 60 | 596.869 | 348.717 | 532.862 | 5000 | 0.4419 |
| 7 | 74.9998 | 400 | 60 | 438.3 | 324.987 | 575.681 | 5000 | 1.2152 |
| 8 | 38.9804 | 356.259 | 60 | 582.463 | 340.447 | 568.946 | 5000 | 0.2336 |

*3.4. Extraction Kinetics*

Unlike the previous experiments in which extraction was performed immediately after exposure to PEF, this section concerns the analysis of the extraction kinetics of each compound during the 24-h maceration period. The results plotted in Figure 6 were obtained with the optimal values obtained previously (T = 39 μs, n = 356, and [C] = 60%). The effect of PEF treatment on the phenol compound was significant, and the effect was more pronounced on total phenols.

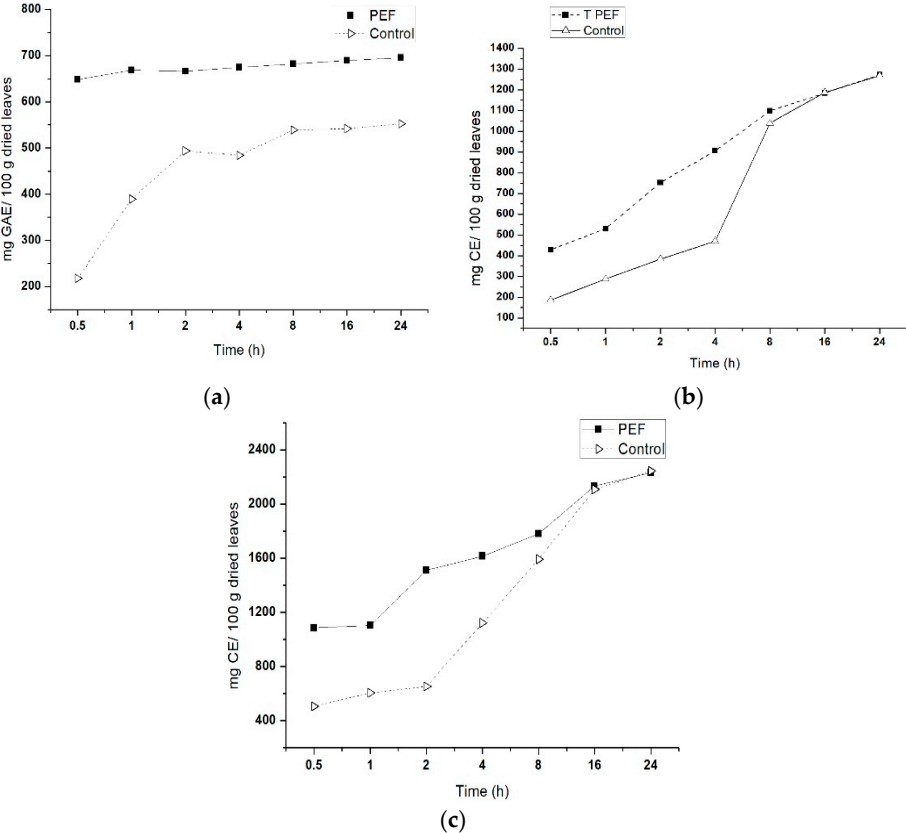

**Figure 6.** Extraction kinetics of the three compounds. (**a**) Total phenols; (**b**) Total flavonoids; (**c**) Condensed tannins.

The PEF treatment allowed the extraction of approximately 680 mg of total phenol in 30 min, whereas for the control sample, the maximum amount of approximately 520 mg was obtained after 24 h. Regarding flavonoids and tannins, the PEF treatment produced a greater amount of extracted product during the first 4 h of maceration. However, after the 24-h maceration period, the yields from the treated and control samples were almost similar.

The influence of PEF was significant in these experiments for all three phenolic compounds compared with that observed in the preliminary experiments. This difference was observed because extraction was performed after a maceration period of 30 min (for the first measurement point), whereas extraction in the preliminary experiments was performed immediately after PEF treatment. On the other hand, the *Eucalyptus* leaves used in these experiments were harvested 2 months after those used in the preliminary experiments.

### 4. Conclusions

This study was conducted to analyse the PEF effect on the yields of phenolic substances extracted from *Eucalyptus* leaves. The study confirmed the significant effect of PEF on the quantity of extracted phenols, flavonoids, and tannins, but it especially revealed the effect on extraction speed, particularly for phenols. The effect of PEF was clearly significant for total phenols; a 25% higher quantity was obtained in a few minutes compared with the equivalent quantity of the untreated sample obtained in 24 h. This fraction contained a large number of phenolic compounds including simple phenols, phenolic acids, and flavonoids. The method of determination used in this study allowed the detection of all these compounds and is not very specific as it is the case for the determination of flavonoids and condensed tannins. Our study showed that PEF facilitates the extraction of the global content of phenolic compounds, which is a promising finding because these compounds serve as metabolites having a high medicinal value.

**Author Contributions:** Literature review, M.N.T. and A.B.; Methodology, A.S., Y.B., M.N.T. and A.B.; Experimental design methodology, A.S. and Y.B.; Experimental work, M.N.T., A.B. and F.T.; Formal analysis, F.T., M.A.B. and A.T.; Writing—review and editing, M.N.T., A.B. and A.T.; Supervision, F.T. and M.A.B.; Funding acquisition, F.T. and M.A.B. All authors have read and agreed to the published version of the manuscript.

**Funding:** This research was funded by by the General Directorate of Algerian Scientific Research (DGRSDT), grant number 362.

**Institutional Review Board Statement:** Not applicable.

**Informed Consent Statement:** Not applicable.

**Data Availability Statement:** Not applicable.

**Acknowledgments:** This study was conducted under the framework of 'Projet Impact Socio-Economique' Contract N° 362 funded by the General Directorate of Algerian Scientific Research (DGRSDT).

**Conflicts of Interest:** The authors declare no conflict of interest.

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
