# Peer review of "Intensification of the Extraction Yield of Eucalyptus globulus Phenolic Compounds with Pulsed Electric Field"

_applsci, doi:10.3390/app12199455_

Round 1

Reviewer 1 Report

The manuscript Intensification of the extraction yield of Eucalyptus globulus phenolic compounds assisted by pulsed electric field submitted to Applied Science - Manuscript Number: applsci-1824803 describes the applicability of PEF on the extraction of  total phenols, total flavonoids and condensed tannins from Eucalyptus globulus leaves, with special regard to yield enhancements and a reduction of processing time.

 Some remarks have to be taken into account by the authors:

 -        How is the reproducibility of the used procedures? This should be evidenced by the repeating of the preparation experiments and characterization and evaluation of the results?

-       - How many samples of one type (or the number of replicates of the experiment) were used in the studies?

    -  The obtained results should be compared to previous reports from literature.

Author Response

See pleased the attached file

Reviewer 2 Report

Dear authors, I have the following comments that might help you improve your manuscript.

-In the introduction section: Did previous works with PEF show an effect on the phenolic constituents of the samples?

-Plant material: was the working material identified by a botanist?

-You mention in the introduction that you optimized the extraction using RSM but you do not specify this in the statistical design. Please mention the optimization design.

-I suggest you to adjust the figure formats.

-Section 3.1 seems to appear out of nowhere, please describe all used methodologies in its respective section.

-Figure 6 must be deleted, and its information presented as a Table.

-The inclusion of the LC-MS analysis would greatly enrich your manuscript as it would allow you to compare also the effect of the extraction conditions on the phenolic profile.

I hope my recommendations are helpful.

Best wishes,

Author Response

See pleased the attached file

Reviewer 3 Report

1. Please update some references, because the publication time is too long.

2. The author optimized the extraction process of Eucalyptus globulus. However, the activity of the extract of Eucalyptus globulus is not clear. 

3. English is very poor, and written language is the suitable choice.

4. The introduction of  the extract technology  of Eucalyptus globulus should be discussed in more detail.

Author Response

See pleased the attached file

Round 2

Reviewer 2 Report

Hi dear authors,

Thank you for your response to our comments, as you know as reviewers we hope that the exchange of ideas enriches our works.

In point 2, regarding the plant material, please add the information you mention in the response to the manuscript.

Tables 1 and 2 need editing, table 2 is composed by two sets of tables with different columns.

In some places of the manuscript when you mention "experimental design methodology", do you mean the "optimization design methodology"? If so, please correct this.
